



# Simulating soil heat transfer with excess ice, erosion and deposition, guaranteed energy conservation, adaptive mesh refinement, and accurate spin-up (FreeThawXice1D)

Niccolò Tubini[1] and Stephan Gruber[1]

[1]Department of Geography and Environmental Studies, Carleton University, Ottawa, ON, K1S 5B6, Canada

**Correspondence:** Niccolò Tubini (tubini.niccolo@gmail.com)

**Abstract.** Thawing permafrost with excess ground ice can cause surface subsidence, damage to infrastructure, and long-term environmental changes. Accurate simulation of the depth, timing, and magnitude of excess-ice melt is important but remains difficult due to nonlinear phase-change dynamics, limitations in model resolution, and the computational challenges of conserving energy over long timescales. Many models blur key features such as the depth of thaw fronts, leading to uncertainty
in assessing related hazards.

To address this, we developed a one-dimensional heat-transfer model that can accurately represent the melting of excess ice along with changes in soil geometry due to erosion or deposition. Innovations include adaptive mesh refinement around the melting point, separate treatment of excess and pore ice, and a two-step spin-up routine that ensures thermal equilibrium in deep profiles. The model uses a semi-implicit scheme with a nested Newton solver that guarantees energy conservation and
convergence at large time steps.

Test cases show that model resolution and regridding influence the timing and magnitude of surface subsidence and thaw penetration. Tracking permafrost change requires representing the dynamic ground-surface elevation and reporting measurements either relative to it or as heights above a fixed datum. Cases with erosion or deposition demonstrate that even modest changes to surface geometry can alter subsurface thermal regimes, and delay or accelerate ice melt.

FreeThawXice1D provides a reliable and extensible tool for research, model testing, and scenario analysis. Its robust numerics and accuracy make it suitable for improving the realism of long-term permafrost simulations and supporting adaptation decisions in cold regions.

## 1  Introduction

Ground ice affects a wide range of phenomena in cold regions (Hock et al., 2019; Meredith et al., 2019). Changes in excess ice
(in excess of the amount of water that the ground would hold under natural, saturated, unfrozen conditions) content result in surface deformation (Shiklomanov et al., 2013; Kokelj and Jorgenson, 2013) and can affect engineered structures (Streletskiy et al., 2019; Suter et al., 2019), ecosystems (Zona et al., 2011) and biogeochemistry (Ekici et al., 2019). These impacts make changes in ground ice relevant for diverse groups of people in cold regions and globally (Gruber et al., 2023).



Quantifying transient depth profiles of ground-ice content and temperature together is of broad relevance. International permafrost monitoring (Sessa and Dolman, 2008) is focused on the thermal state of permafrost and active-layer thickness, both of which are confounded by the loss of excess ground ice. Ground-ice loss and surface movement (O'Neill et al., 2019; Cai et al., 2020; Gruber, 2020) are relevant additional metrics of permafrost thaw to complement the established variables used in the majority of simulations today (e.g., Garnello et al., 2021). Understanding the spatial differentiation of ground-ice profiles in a landscape represents an intersection of the ice that was in the ground at some point in time, and the ground temperature evolution that melted this ice since then (Burn, 1997; Subedi et al., 2020). Simulating past patterns of thaw depth will, therefore, likely become an important complement for geologic investigations of ground-ice distribution because it can identify locations and depth ranges, where excess ice is no longer to be expected.

The timing and magnitude of impacts from permafrost thaw are strongly controlled by the depth at which excess ice occurs. Key challenges in quantifying the evolution of excess ice in soil profiles and concomitant hazards, therefore, relate to the availability of (a) data representing the distribution of ice in the ground, suitable as input for simulations, and (b) simulation tools with the ability to accurately represent the timing of its melt.

The deposition and erosion of material at the ground surface, rapidly or gradually, can contribute to the formation of excess ice (Gruber and Haeberli, 2009; Abramov et al., 2008) or to exhumation via denudation (Basisty and Buiskikh, 1998), affecting the ground thermal regime. Engineering analogues with deposition or erosion of surface materials include the storage of waste-rock or tailings (Knutsson et al., 2018) and pre-thaw by stripping parts of the surface layer (Linell, 1973; Esch, 1982).

A range of models already represent excess ice: Land-surface models such as the Northern Ecosystem Soil Temperature (NEST) model (Zhang et al., 2003), the Community Land Model (CLM, Lee et al., 2014), and CryoGrid (Westermann et al., 2016) represent excess-ice melt as well as heat and water transport in the soil, the surface energy balance, the seasonal snow pack and the vegetation canopy. CryoGrid now includes the ability to represent the formation of excess ice by segregation (Aga et al., 2023). These models are suitable for simulating many locations over years to centuries.

Engineering models represent the mechanics of excess ice dynamics in more detail and sometimes in multiple dimensions (e.g., Hwang, 1976; Nishimura et al., 2009; Yu et al., 2020), while being specific to particular use cases and local configurations. Other models allow the explicit or parameterized simulation of fine-scale lateral patterns of excess-ice loss (Jan et al., 2018; Painter et al., 2023) as a means to simulate thermokarst.

This list of models is not complete, but it illustrates the diversity of previous work. Each of these models has specific strengths and simplifications reflecting their intended purpose, navigating trade-offs with respect to computational demand, accuracy, and the detail of process representation.

The effects of these simplifications, however, are not known quantitatively. Many simulation tools erroneously broaden the thawing zone with depth, blurring the timing and magnitude of ground ice melt. This can be caused by: (1) Artifacts from numerical solvers (e.g., the Decoupled Energy Conservation Parametrization, short DECP, Tubini et al., 2021), (2) coarse grid resolution with depth or the assignment of average ice content for large vertical sections, (3) numerical solvers that may be unable to converge when faced with the steep enthalpy function of pure (excess) ice and either introduce errors or require the use of overly gentle regularizations or soil freezing characteristic curves (SFCCs).





A model that can accurately reflect the depth-time relationship of ground-ice loss over years to millenia is desirable and can help elucidate the impacts that necessary trade-offs in models have on their ability to represent the depth, timing, and magnitude of ground-ice melt. Moreover, the duration and depth of the simulated domain are connected because of the need for sufficient depth to reflect transient changes, the length of spin-up, and the overall simulation time. The investigation of ground-ice loss over long time periods, therefore, requires numerical efficiency and accuracy.

Extending previous work (Tubini et al., 2021), this contribution describes and demonstrates a model with six specific capabilities: (1) The discretization of the soil column adjusts as excess ice is melted or formed or when material is added or removed at the surface. The desired subsurface quantities are reported relative to the moving ground surface and relative to a fixed datum. (2) The thermophysical ground characteristics change in response to temperature and excess ice content. (3) The computational mesh is adaptively refined where strong phase change is likely to occur. (4) The freezing characteristic curves of soil and of excess ice are considered separately in each volume element. (5) The spin-up of soil profiles into equilibrium accommodates the interacting changes in temperature, geometry and ice content. (6) Energy conservation and model convergence at large time steps are guaranteed.

The capabilities are important for accurately simulating ground-ice dynamics over hundreds to many thousands of years, even in the presence of the extremely steep enthalpy function typical for excess (pure) ice.

## 2 Model description

The energy transfer in the ground is governed by heat conduction and phase change, while the soil water balance is not accounted for.

### 2.1 Subsurface heat transfer

The soil thermal regime is described by the enthalpy equation

$$\frac{\partial h(T)}{\partial t} = \frac{\partial}{\partial z} \left( \lambda(T) \frac{\partial T}{\partial z} \right) + S \tag{1}$$

where $h(T)$ is the volumetric enthalpy (J m$^{-3}$), $\lambda(T)$ is the thermal conductivity (W m$^{-1}$ K$^{-1}$), $T$ is temperature (K), $t$ is time (s), and $z$ (m) is the vertical coordinate assumed positive upward, and $S$ is the source-sink term.

### 2.2 State equation

Since enthalpy is an extensive variable, the volumetric enthalpy is defined as the sum of the enthalpy of saturated soil and the enthalpy of excess ice. The volumetric enthalpy of the soil without excess ice can be calculated as the sum of the internal energy of the soil particles, ice and liquid water content

$$h_{soil}(T) = h_{sp}(T) + h_w(T) + h_i(T) \tag{2}$$



$$h_{sp}(T) = c_{sp}\rho_{sp}(1 - \theta_s)(T - T_{ref}) \tag{3}$$

$$h_i(T) = c_i\rho_i\theta_i(T)(T - T_{ref}) \tag{4}$$

$$h_w(T) = c_w\rho_w\theta_w(T)(T - T_{ref}) + \rho_w\theta_w(T)l_f \tag{5}$$

where $c$ is specific heat capacity (J kg$^{-1}$ K$^{-1}$); $\rho$ is density (kg m$^{-3}$); subscripts $_{soil}$, $_{sp}$, $_w$, and $_i$ refer to, respectively, the bulk soil, soil particles, liquid water, and ice; $\theta_w(T)$ is the unfrozen water content, $\theta_i(T)$ is the ice content, and $l_f$ is the specific latent heat of fusion of water. $T_{ref}$ is a reference temperature that is usually equal to the melting temperature of bulk water at standard atmospheric pressure. The liquid water content is modeled using the SFCC presented by Dall'Amico et al. (2011), but other parameterizations can be included with minimal effort.

The volumetric enthalpy of excess ice is

$$h_{ei}(T) = \begin{cases} \rho_i c_i(T - T_m) & \text{, if } T < T_m \\ \rho_w c_w(T - T_m) + \rho_w l_f & \text{, otherwise,} \end{cases} \tag{6}$$

where $T_m$ is the melting temperature at atmospheric pressure. The singularity at $T = T_m$ necessitates a linearization of Eq. (6) at $T = T_m$

$$h_{ei}(T) = \begin{cases} \rho_i c_i(T - T_m) & \text{, if } T < T_m - \epsilon \\ \rho_w c_w(T - T_m) + \rho_w l_f & \text{, if } T > T_m \\ \rho_i c_i(-\epsilon) + \Delta h(T - (T_m - \epsilon)) & \text{, otherwise,} \end{cases} \tag{7}$$

where $\epsilon$ (K) is a parameter defining the temperature range over which the phase change of excess ice occurs, and $\Delta h = \dfrac{\rho_w l_f - \rho_i c_i(-\epsilon)}{\epsilon}$. While necessary numerically, this linearization also has a physical basis as even in water without soil, phase change occurs over a finite range in temperature (Nye and Frank, 1973; Langham, 1974).

Finally, the enthalpy of a control volume $V = V_{soil} + V_{ei}$ (m$^3$) can be computed as

$$H(T) = h_{soil}(T)V_{soil} + h_{ei}(T)V_{ei} \tag{8}$$

## 2.3 Time integration

Eq. (1) is solved numerically by using a semi-implicit finite volume scheme and and the nested Newton algorithm (NCZ) (Casulli and Zanolli, 2010) is used to solve the resulting nonlinear system of equations (Tubini et al., 2021). This method is unconditionally stable and energy conservation is guaranteed for any spatial and temporal discretization.

## 2.4 Soil freezing characteristic curve and thermal conductivity

Equation (1) must be completed with two constitutive relationships describing the thermal conductivity $\lambda(T)$ and the SFCC $\theta_w(T)$.



A peculiar feature of frozen soils is that liquid water and pore ice can coexists in a broad range of temperatures. This is due to four factors that can lower the melting point of water in porous media compared to that of bulk water at standard pressure: the Gibbs–Thomson effect causes pre-melting in pores due to curved ice-water interfaces, pre-melting on charged mineral surfaces, the presence of solutes, and hydrostatic pressure (Watanabe and Mizoguchi, 2002; Rempel et al., 2004; Christoffersen and Tulaczyk, 2003). Additionally, delayed nucleation can lower the temperature of freezing further with respect to the melting temperature.

The relationship between the content of unfrozen water and temperature (the SFCC, e.g., Devoie et al., 2022; Morgenstern and Anderson, 1973; Koopmans and Miller, 1966) and its parameterization is of critical importance when modeling frozen soil. In fact, the soil bulk thermal conductivity and heat capacity, depend on the thermal properties of its constituents and their abundance. Several SFCC models have been presented in the literature (Kurylyk and Watanabe, 2013) and can be divided in two groups (Ren et al., 2018). Empirical models are based on empirical curve-fitting for obtaining functional relationships (e.g., Anderson and Tice, 1972; McKenzie et al., 2007; Kozlowski, 2007). Models based on the soil water retention curve use the Clausius-Clapeyron equation to derive SFCCs (Shoop and Bigl, 1997; Dall'Amico et al., 2011; Sheshukov and Nieber, 2011) that are a function of both temperature and unfrozen water pressure (saturation) (Kurylyk and Watanabe, 2013). In FreeThawXice-1D, the Dall'Amico et al. (2011) model, and the McKenzie et al. (2007) model are implemented.

In Eq. (1) the thermal conductivity, $\lambda(T)$, is a combination of the soil thermal conductivity and that of the excess ice, when it is present. The arithmetic mean, the harmonic mean, or the geometric mean can be chosen as mixing models for this in response to differing cryostructures (Fig. 1B,C). While many mixing model exist, these classical mixing laws approximate a series configuration with the harmonic mean (lower bound, e.g., excess ice layered orthogonal to heat-flow direction as in Fig. 1B), a parallel arrangement with the arithmetic mean (upper bound for bulk conductivity), and an idealized random distribution with the geometric mean (e.g., particles suspended in ice, as in Fig. 1C) (cf. Dong et al., 2015).

The thermal conductivity of soil is a combination of the thermal conductivity of its constituents: soil particles, pore water, and pore ice. Many models of soil thermal conductivity exist (Dai et al., 2019; Dong et al., 2015) and in FreeThawXice-1D, the Johansen model (Johansen, 1977) and the Cosenza model (Cosenza et al., 2003) are implemented.

There is no agreement on the best model for the SFCC and bulk thermal conductivity (Kurylyk and Watanabe, 2013; Dai et al., 2019; Wang et al., 2021) even though they are critical for modelling heat transfer in soil and rock (Ochsner et al., 2001; Nicolsky and Romanovsky, 2018; Amankwah et al., 2021; Dai et al., 2019; Wang et al., 2021).

To accommodate new models and their testing, FreeThawXice1D allows including new SFCC and thermal conductivity models easily and neatly (Clark et al., 2015). FreeThawXice1D has been implemented following the Object Oriented Paradigm (OOP) approach. Specifically, the classes representing the SFCC and thermal conductivity models have been implemented by adopting the so-called Simple Factory strategy pattern (Gamma et al., 1995; Freeman et al., 2008; Tubini and Rigon, 2022). At the base of this pattern there is a so-called abstract class that exposes the methods, whereas their concrete implementation is left to the concrete classes. In this manner, developers can include new SFCC and thermal conductivity models by writing new classes that must fulfil the interface with the abstract class without having to read, understand, and modify existing tested code, which is usually error-prone.





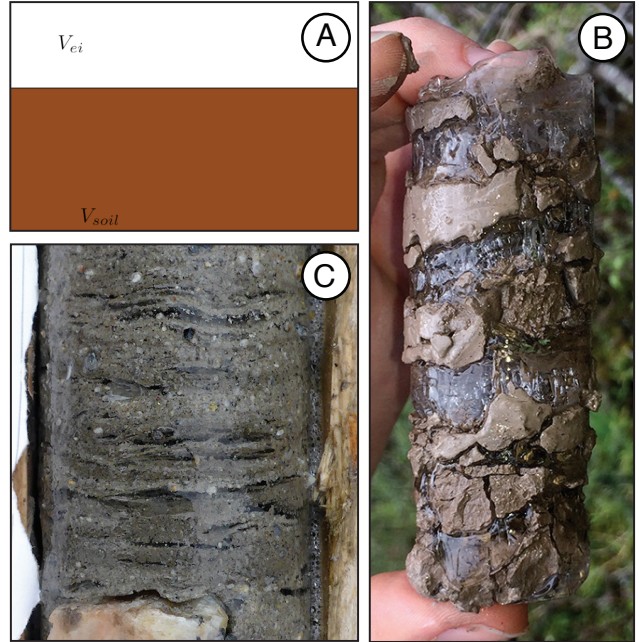

**Figure 1.** (A) The scheme of a control volume containing saturated soil (brown), $V_{soil}$, and excess ice (white), $V_{ei}$. The enthalpy of a control volume is the sum of the enthalpy of saturated soil and the enthalpy of excess ice. (B) Drill core showing clear layers of segregated excess ice and of frozen silty clay. In FreeThawXice1D, several layers together may be represented by one control volume as shown in (A), or alternatively, individual layers may be described as saturated soil only and ice only. Excess ice can also be more difficult to recognize visually, as shown by the drill core in panel (C) that has 45% excess ice (Subedi et al., 2020).

## 2.5 Excess ice

Excess-ice melting is based on the ground thermal regime, and excess-ice content and geometry are directly linked. A stratigraphy that may include excess ice is defined as an initial condition of the simulation and at each time step, the energy equation is solved numerically, and the excess-ice volume adjusted, consequently. At the end of each time step, the computational grid is adjusted to the melted or newly formed excess ice volume.

The melted volume is computed according to the difference in the enthalpy of excess from one time step ($t^n$) to the next ($t^{n+1}$). Specifically, it is proportional to the ratio between the difference of latent heat and the latent heat required to melt the entire excess ice volume (Fig. 2). Since water movement in the soil column is not represented, the meltwater and the associated amount of latent heat are removed directly from the simulation domain (cf., Gambino et al., 2019).

## 2.6 Surface boundary conditions

From the numerical point of view, the soil surface can be specified as either a temperature ('Dirichlet') or a heat flux ('Neumann') boundary condition. To represent the 'buffer layer', the combined effects of snow and vegetation (Vincent et al., 2017),





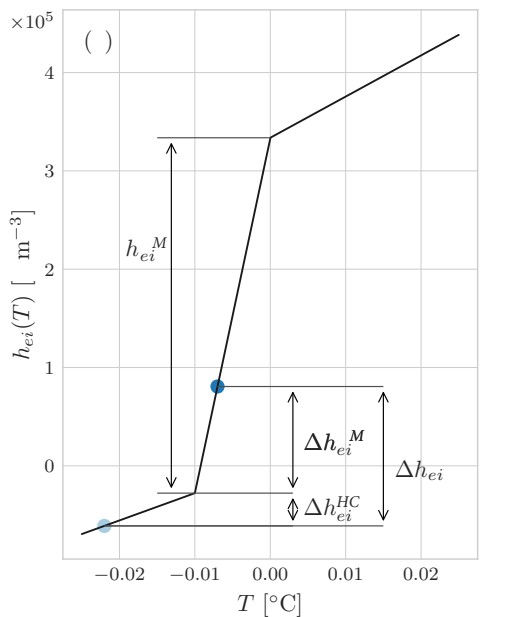
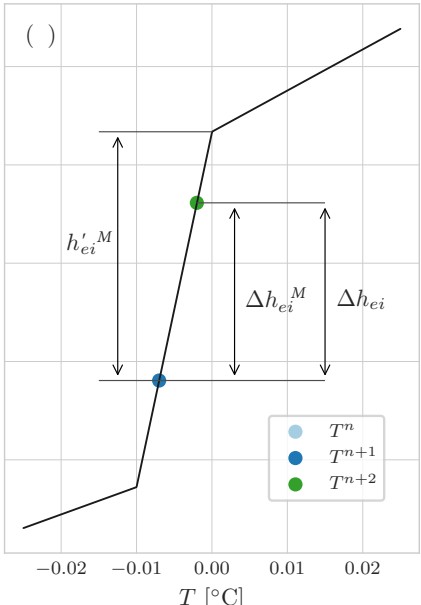

**Figure 2.** Melting of excess ice: the volumetric enthalpy of ice, $h_{ei}$, has been linearized with Eq. (7) to avoid the singularity at $0\,°$C. In panel (a), temperature evolves from $T^n$ to $T^{n+1}$ where $n$ is a generic time step of the simulation. Passing from time $n$ to $n+1$, the proportion of excess-ice volume melted is equal to ratio between $\Delta h_{ei}^{LM}$ and $h_{ei}^{LM}$, with superscripts $^{LM}$ and $^{HC}$ indicating the portions of linearized melt and only heat capacity, respectively. In panel (b), temperature increases from $T^{n+1}$ to $T^{n+2}$ so the corresponding proportion of melted volume is the ratio of $\Delta h_{ei}^{LH}$ and $h_{ei}^{'LM}$, which represents the amount of heat required to complete the melting of the remaining excess ice volume.

in a parsimonious way, a further option is included. The heat flux at the ground surface is modelled as

$$\mathcal{F} = -\mathcal{H}(T_{air} - T_{soil}) \tag{9}$$

where $\mathcal{F}$ is the heat flux [Wm$^{-2}$], $\mathcal{H}$ is the heat-transfer coefficient [Wm$^{-2}$K$^{-1}$], $T_{air}$ [K] is the air temperature, and $T_{soil}$ [K] is the temperature of the soil surface. For a medium of finite thickness such as a snow pack, $\mathcal{H}$ would be derived as a thermal transmittance. In equilibrium, it is the inverse of thermal resistance and with oscillating temperature conditions, it may

be reduced due to temporary storage and release of heat by the medium characterised.

An additional condition is required to force the ground surface to remain at or below $0\,°$C while a snow cover is prescribed, representing the latent heat effect of snow in a simplified manner. First, the heat equation is solved by using the heat-transfer coefficient. Then, if the new temperature of the uppermost control volume is above $0\,°$C and a snow cover is present, the surface boundary condition is switched to Dirichlet and the time step is solved again with a boundary condition of $0\,°$C (Fig.

A1 and A2).





## 2.7 Regridding

Simulating excess ice dynamics, and material deposition and erosion at the ground surface, require a mobile computational grid. Here, we do not consider the soil mechanics (see Dumais and Konrad, 2018), and geometry and excess ice as well as surface erosion and deposition are directly linked. The thickness of individual soil layers, and consequentially of the entire soil

column, is reduced or increased proportional to the melted or newly formed volume of excess ice. Similarly, the deposition or erosion at the soil surface determine changes in the surface elevation. These changes of the computational domain require a routine that prevents control volumes from becoming either too thick (splitting layers) or too thin (merging layers) as this would incur a loss of accuracy or unnecessary computational cost. Additionally, a good resolution close to the zero-degree isotherm is more important to represent the phase change in our numerical scheme than the temporal resolution (Tubini et al.,

180    2021).

When merging control volumes, only those with identical soil parameters (except for excess-ice content) are combined to prevent changing the soil stratigraphy. The merging conserves soil and excess-ice volume as well as internal energy. Combining volumes with different temperatures determines a new thermal equilibrium that is instantaneously reached. To avoid artificial changes to the position of the zero-isotherms, it is only possible to merge either thawed control volumes or frozen ones.

However, because merging control volumes may introduce an artificial freezing/thawing of pore ice and excess ice, it is possible to specify a temperature threshold, $\hat{T} < 0°\mathrm{C}$, such that in the interval $[\hat{T},0]°\mathrm{C}$ merging is not done.

When splitting, soil volume and excess ice are equally divided among the new control volumes that inherit their soil parameters from the parent. Volume and internal energy are conserved.

As grid size determines how accurately the evolution of the zero-isotherms can be captured (Tubini et al., 2021), the dis-

cretization can be adjusted in proximity of the zero isotherms. This is done by specifying a temperature interval around $0 °\mathrm{C}$ and specifying the desired size of the control volumes. Away from the zero isotherm, the control volumes are merged to reduce computational cost.

## 2.8 Spin-up

Spin-up (Noetzli and Gruber, 2009; Ross et al., 2021) aims to achieve a stable temperature distribution consistent with site

conditions, and independent of the initial condition prescribed. As such, most transient simulations require an equilibrium period that is simulated many times consecutively for spin-up before the transient period of interest is simulated. This generally requires long simulation periods because phase change in the soil increases the amount of heat that needs to be transferred though the soil column, while also keeping temperature gradients, which drive heat conduction, low. Using borehole observations as initial conditions is difficult because their thermal regime may differ from that of the model, leading to a mixed signal

of diminishing differences between observation and model that will confound the transient model response desired.

Accurately simulating long-term permafrost changes has been shown to require deep, often several tens to hundreds of metres, soil columns (Delisle, 2007; Lawrence et al., 2012; Sapriza-Azuri et al., 2018). Depth, however, increases thermal memory and the duration and computational cost for simple spin-up, understood here as repeated simulation of the entire soil





column for the equilibration period. Because inaccurate spin-up influences the results of transient simulations, an effective

spin-up routine is an important feature for permafrost models.

FreeThawXice1D includes a two-step procedure (Gubler et al., 2013) to efficiently and accurately spin-up deep permafrost columns. It extends and improves the routine included in GEOtop (Endrizzi et al., 2014) to account for the presence of excess ice and variable surface elevation.

First, the shallow part of the soil column is spun up for the equilibration period, finding the ground thermal regime resulting

from seasonal and inter-annual variation at the surface and the thermal offset. A thickness of 10–20 m is usually sufficient to include the depth of significant seasonal and inter-annual variation.

Second, deeper ground temperatures are computed, which are controlled by the geothermal heat flux and ground thermal conductivity and only show negligible temporal variation during the equilibration period. This is done by solving the stationary heat equation:

$$\lambda(T)\frac{\partial T}{\partial z} = \mathcal{G} \tag{10}$$

where $\mathcal{G}$ is the geothermal heat flux ($\mathrm{Wm^{-2}}$). Starting with the mean equilibrated temperature of the lowest node of the shallow domain, layer-by-layer downward iteration is required since the thermal conductivity is a function of temperature. Once the temperature profile has been computed for the entire soil column the need for grid refinement close to the zero isotherm is checked. If needed, the stationary heat equation is solved again with a locally refined grid. This procedure reduces

the computational burden of spin-up since the non-stationary heat equation must be solved only for the shallow part of the soil column and for a shorter duration than if equilibrating a much deeper profile.

In the absence of suitable analytical solutions or other true reference values, judging the completeness of spin-up is inherently difficult. Because the release of latent heat at depth affects the difficulty of spin-up strongly, the difference between spin-up from a uniform temperature of 1 °C and of -1 °C is used (following Gubler et al., 2013) to measure, in a setting without excess

ice, how successfully the procedure can establish independence from initial conditions.

Excess ice complicates spin-up by its potential for changing the geometry of the soil column. The depth of the shallow domain for spin-up must account for excess-ice losses and an additional, preliminary spin-up step can be used to equilibrate the surface elevation with the boundary conditions when the potential for large subsidence exists. Preliminary spin-up reveals an appropriate surface elevation and near-surface distribution of ground ice by simulating the entire soil column during the

equilibrium period several times. Based on this, the required extent of the shallow domain can be chosen. After this step, we proceed with the two-step procedure for spin-up: We perform shallow spin-up to get the equilibrium temperature profile. Finally, we compute temperatures at depth as described above, accounting for any melting excess ice at the base of permafrost requiring adjustment of the grid geometry.

## 2.9   Input and output

The input required to run a simulation can be grouped into three categories:



1. Computational grid data is comprised of the geometrical data for the computational domain, the stratigraphy of the soil column describing the vertical distribution of ground materials, their excess-ice content (Castagner et al., 2023), and the initial temperature profile. This information is specified by the user through four different .csv files and then processed in a Python script to define the computational grid and write the NetCDF files that FreeThawXice1D reads.

2. Time series specify the boundary conditions: the presence of a snow cover and the amount of erosion and deposition at the soil surface. Time series use a OMS3-compliant (David, 2010) .csv format.

3. The simulation parameters, such as the start date and end date of the simulation, time step size, and the file paths, are specified by the user in the OMS3 .sim file.

Outputs are stored in a NetCDF-3format (Unidata, 2021). NetCDF is a self-describing portable data format developed and maintained by UCAR Unidata. It is commonly used in atmospheric and oceanic science and there is an ever-growing number of tools for processing and visualization. Use of NetCDF for permafrost data is increasing (Brown et al., 2024; Brown, 2022).

Since simulation output can be very large, we designed the output workflow following the approach presented by Tubini and Rigon (2022). In this manner the user has the opportunity to chose among different output strategies and eventually create a new one that can differ from the others for the variables saved or for the file format. Currently, available output strategies consist in: saving all the simulation variables, saving aggregated variables, variables at different depth below the ground surface or height above a datum, saving only the mean temperature profile for spin-up purpose, or saving a back up of the last simulation time step to enable resuming a simulation.

## 3 Demonstration with test cases

In this section, we show test cases demonstrating the capabilities of FreeThawXice1D and highlight interesting transient effects in permafrost soils with excess ice that can be revealed with accurate long-term simulations. The novel surface boundary condition is parameterized and tested in Brown and Gruber (2025). Where necessary, detailed additional visualisations are contained in the Appendix.

### 3.1 The importance and effectiveness of regridding

To demonstrate the effectiveness of controlling the computational grid we consider a $10$ m soil column in which the upper $1.5$ m contain excess ice, $80\%$ in volume. As initial condition for the temperature we assume that it varies linearly from $+3$ °C at the bottom to $-2$ °C at the top. As boundary conditions at the bottom we use a constant heat flux, $0.05$ W m$^{-2}$, and at the surface, we impose a sinuoidal Dirichlet boundary condition with mean temperature of $-3$ [°C] and an annual amplitude of $10$ [°C]. With this configuration, we have a zero-isotherm close to the soil surface and a second one deeper, around $-3.5$ m depth from the soil surface.

We compare aggregated metrics of surface elevation and the position of the zero-isotherms for these simulations: (1) A reference simulations without regridding that has 10,000 elements, each one $0.001$ m thick. (2) Three simulations with a





coarse grid of 200 elements and a $\pm 0.2$ °C temperature interval around 0 °C to refine with desired control volume sizes of 0.01 m, 0.005 m, and 0.001 m. (3) Three simulations with a coarse grid of 200 elements and a $\pm 0.5$ °C temperature interval around 0 °C to refine with desired control volume sizes of 0.01 m, 0.005 m, and 0.001 m.

These experiments (Fig. 3 and 4) show that regridding to finely discretized control volumes (e.g., 0.001 m) around 0 °C improves simulation results. The width of the temperature interval around 0 °C for regridding does not affect the results but allows to keep the number of grid elements low. In the present case, a 50-fold reduction in the number of layers yielded nearly identical results based on local regridding. Furthermore, Fig. 3b and 4 show how strongly the discretization near the 0 °C isotherm can affect the resulting magnitude and timing of isotherm position and subsidence. Varying layer thickness between

1 mm and 10 mm caused differences in subsidence of up to 25 cm that lasted for more than two decades.

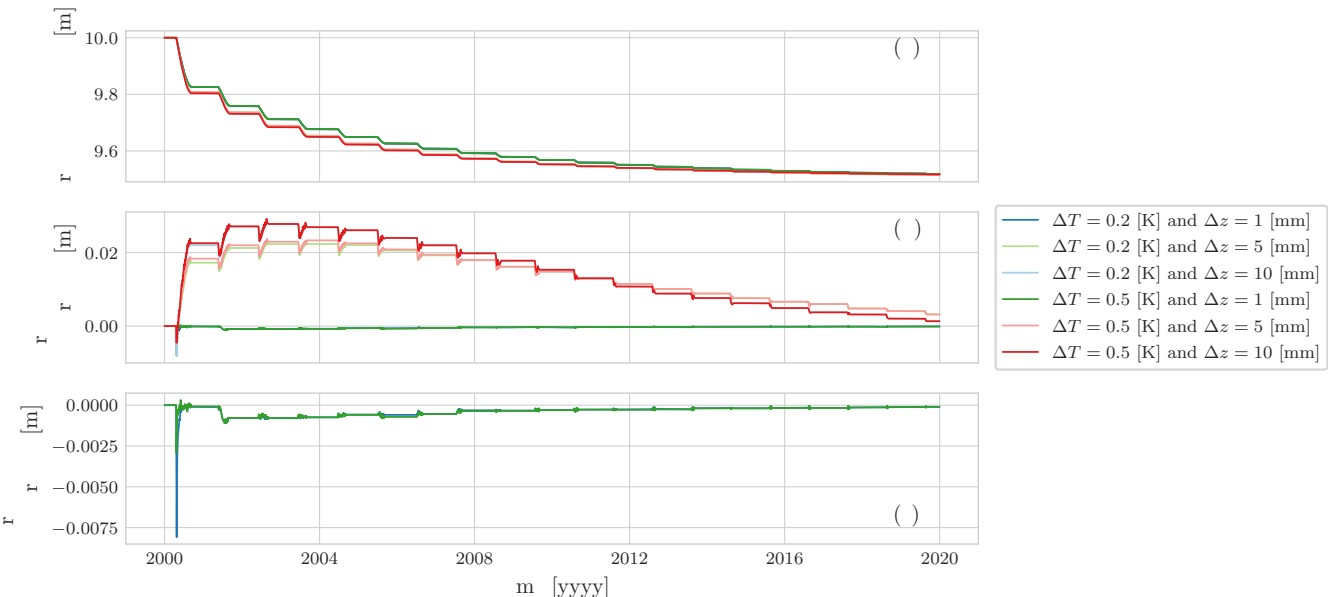

**Figure 3.** The timing and magnitude of simulated subsidence varies with differing regridding parameters. Panel (a) compares changing surface elevation over time. The steps reflect seasonal melt of excess ice in response to the periodic surface temperature. Panel (b) shows the differences with respect to the reference simulation with 10k control volumes of 0.001 m thickness. Panel (c) compares the effect of using a temperature interval of $\pm 0.2$ °C and $\pm 0.5$ °C around 0 °C for refining the control volume size to 0.001 m.





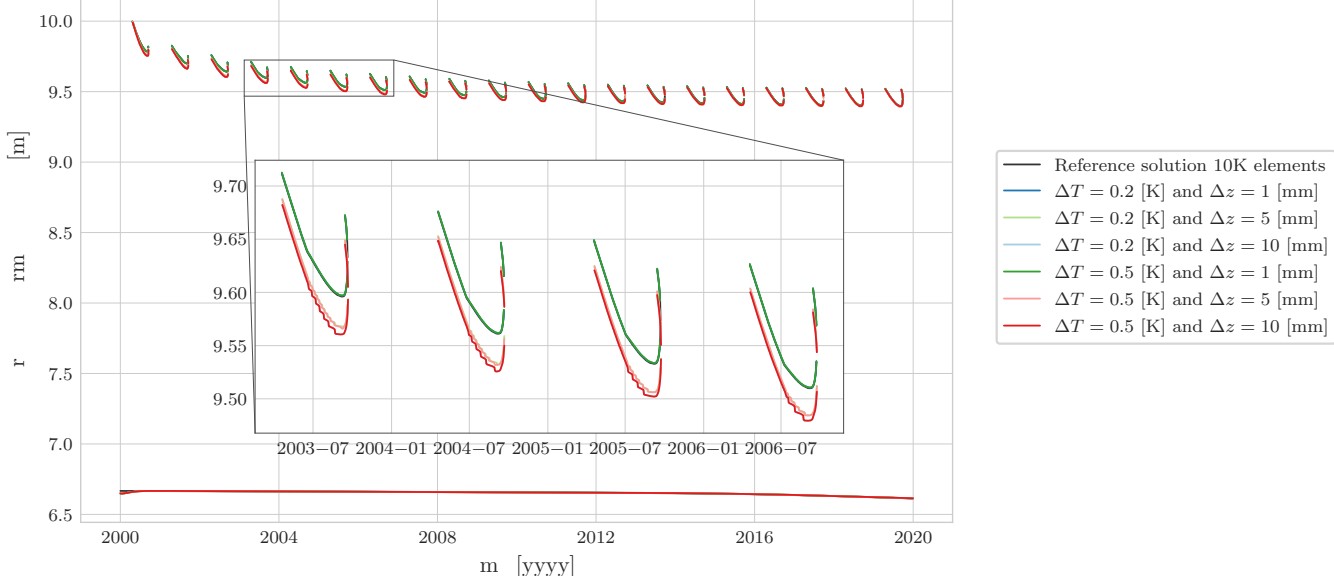

**Figure 4.** The position of zero-isotherms in the ground varies with differing regridding parameters. Control volume sizes of 0.01 m, 0.005 m cause a downward shift in the uppermost zero-isotherm compared to discretizations with 0.001 m.

The grid-dependent differences tend to diminish over time in long simulations when energy conservation is ascertained. However, for simulations that investigate the depth, timing, and magnitude of ground-ice melt, for example, to inform the understanding or anticipation of hazards, grid effects on timing and magnitude need to be taken into consideration.

## 3.2 Case 1: Spin-up without excess ice

We report an example of the two-step spin-up procedure for a uniform, 200 [m] soil column of silt (Tab. 1). The bottom boundary condition is a constant geothermal flux, $0.05 \, \mathrm{W \, m^{-2}}$, and at the surface, we impose a Dirichlet boundary condition as a sine wave with a mean temperature of $-3$ [°C] and an annual amplitude of 10 [°C]. The shallow spin-up is 15 [m] deep with a uniform initial condition $T = -3 \, °\mathrm{C}$.

Fig. (5a) shows the spun-up equilibrium temperature profile. The mean temperature first decreases with depth in the upper
metre due to thermal offset (Riseborough et al., 2008) and then increases in response to the heat flow from below. The geothermal gradient changes near the permafrost base due to changing proportions of ice and water in the ground. Only the duration of spin-up for the shallow part (Fig. 5b) determines the quality even for the 200 m profile and the comparison between the profiles initialized at $-1$ °C and at $+1$ °C demonstrates that high quality can be achieved with the two-step procedure (Tab. 2).

By contrast, simple spin-up from initial conditions of $-1$ °C and $1$ °C for $20,000$ years is still incomplete (Fig. 6), with
different temperature distributions throughout the profile and a difference of 70 m in permafrost thickness. The resulting profiles may visually each be deemed realistic without the comparison to the two-step reference case.





**Table 1.** Ground thermal properties for test cases.

| Test case | Soil type | $C\,\mathrm{KJ\,m^{-3}\,K^{-1}}$ | $\lambda_g\,\mathrm{W\,m^{-2}\,K^{-1}}$ | $\theta_s\,\mathrm{m^3\,m^{-3}}$ | $\theta_r\,\mathrm{m^3\,m^{-3}}$ | $\alpha\,\mathrm{m^{-1}}$ | $n\,-$ |
|---|---|---|---|---|---|---|---|
| 1 | Silt | | 3 | 0.46 | 0.1 | 1.5 | 1.2 |
| 2, 3 | Silt | | 3 | 0.46 | 0.1 | 1.5 | 1.2 |
| | Rock (granite) | $2.447e6$ | 4 | - | - | - | - |

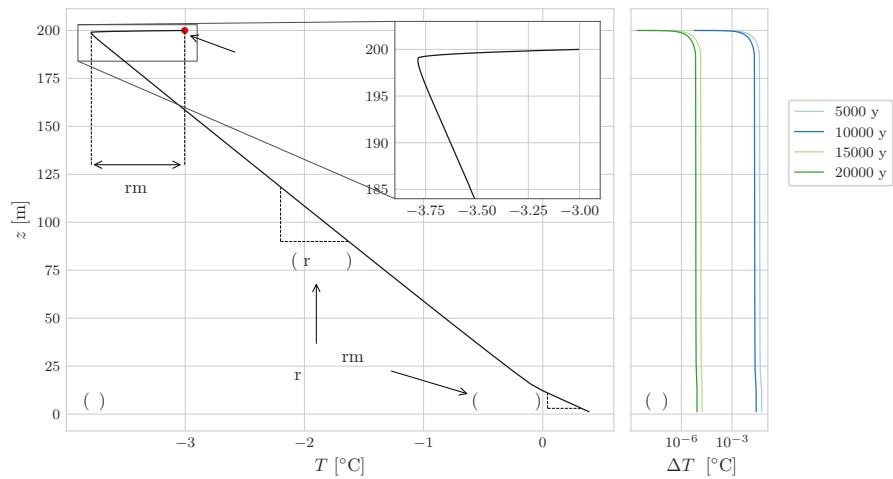

**Figure 5.** Equilibrium mean temperature profile (a) for a silty soil column of 200 m with a periodic Dirichlet boundary condition at the soil surface, mean temperature $-3\,^\circ\mathrm{C}$ and an annual amplitude of $10\,^\circ\mathrm{C}$, and a constant geothermal heat flux of $0.05\,\mathrm{W\,m^{-2}}$ at the bottom. It has been computed using the two step spin-up procedure. The duration of shallow spin-up affects the difference (b) between temperature profiles initialized at $-1\,^\circ\mathrm{C}$ and at $+1\,^\circ\mathrm{C}$. MAGST: mean annual ground surface temperature.

**Table 2.** Maximum absolute differences between equilibrium temperature profiles spun-up with the two-step procedure from $-1\,^\circ\mathrm{C}$ and from $1\,^\circ\mathrm{C}$ for different durations. Profiles are computed for a soil column of silt, 200 m deep.

| Duration years | Max. absolute difference $^\circ\mathrm{C}$ |
|---|---|
| 5,000 | $5.594 \times 10^{-2}$ |
| 10,000 | $2.642 \times 10^{-2}$ |
| 15,000 | $1.675 \times 10^{-5}$ |
| 20,000 | $8.397 \times 10^{-6}$ |





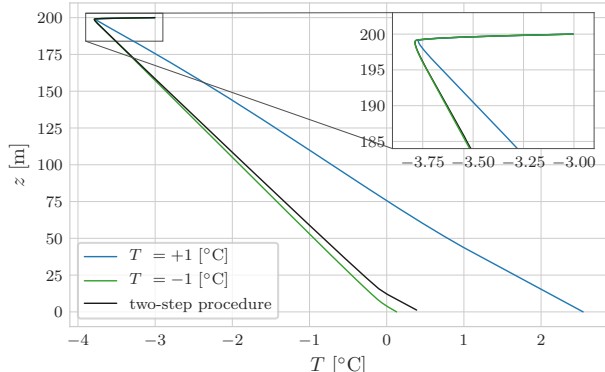

**Figure 6.** Comparison of simple spin-up from initial conditions of $-1\,°C$ and $1\,°C$ for $20,000$ years for a soil column of $200$ m with the accurate two-step spin-up as a reference. Differences increase toward the bottom of the soil column but also exist in the upper part of the soil column where the thermal offset differs. Note the range of about 70m of permafrost thickness between the differing experiments.

The two-step spin-up guarantees equilibration while reducing the computational cost because the non-stationary heat equation must be solved only for the upper part of the soil column, which requires a much shorter duration. As the NCZ algorithm (Tubini et al., 2021) enables large time steps without encountering numerical issues, long and accurate shallow spin-up can be
performed. We have demonstrated the accuracy and importance of two-step spin-up here without excess ice because spin-up from an initial conditions of $1\,°C$ would cause the loss of all excess ice.

### 3.3 Case 2: Spin-up and transient simulation with excess ice

In this second experiment, we consider a $200$ m column where the first $100$ m consist of silt with $80\%$ in volume excess ice, and the remaining $100$ m below consist of compact granite (Tab. 1). To spin up the domain, the bottom boundary condition
is a constant geothermal flux, $0.05\,\mathrm{W\,m^{-2}}$, and at the surface, we impose a periodic Dirichlet boundary condition with mean temperature of $-3\,°C$ and an annual amplitude of $10\,°C$.

Before the two-step procedure, we equilibrate the excess-ice volume with the boundary conditions in a preliminary spin-up. The shallow domain of the soil column needs to be chosen sufficiently deep so that the seasonal variation is contained within the domain even after excess-ice loss (Fig. B1). The resulting soil column has a distribution of near-surface excess-ice,
and possibly an active layer composed of the lag remaining after excess-ice melt, that corresponds with the surface boundary condition. Now, the two-step spin-up, taking into account the expected subsidence when choosing the thickness of the shallow domain, can produce a spun-up soil column (Fig. B2).

The subsequent transient simulation is driven with a surface temperature that for the first $500$ years oscillates around $-3\,°C$ with seasonal fluctuation of $\pm10\,°C$, as the for the spin up. Then, for the subsequent $500$ years, the mean surface temperature
increases to $-1\,°C$, and for the last $1,000$ years the mean surface temperature is $+2\,°C$. At the bottom, a heat flux of $0.05\,\mathrm{Wm^{-2}}$ is prescribed.





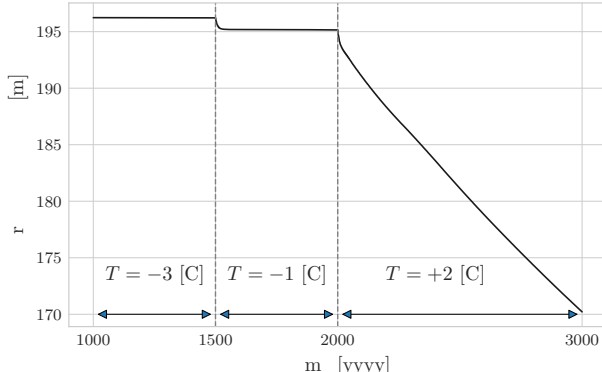

**Figure 7.** Elevation of the ground surface during a transient simulation of $2,000$ years. In the first period, elevation remains stable, indicating no loss of excess ice. Warming in the second period causes loss of excess ice and surface subsidence until a new equilibrium is reached. The third period shows sustained loss of excess ice and subsidence that continues for more than $1,000$ years.

Figure (7) shows the evolution of the ground surface elevation. During the first $500$ years the surface elevation remains constant at $196.24$ m because the thermal regime and excess ice are in equilibrium with the boundary conditions. As the surface warms after model year $1500$, the ground surface subsides by $1.07$ m to a new equilibrium. After model year $2000$, the surface warms again and the column enters a long period of disequilibrium with the forcing boundary conditions, causing the degradation of excess ice at depth and ground subsidence. Initially, subsidence presents a seasonal pattern (Fig. B3) that then becomes subdued as the permafrost table moves to increasing depth.

Complete permafrost degradation and melt of all excess ice in this warming scenario requires more than the $1,000$ years simulated. During the simulation, we can see the rising of the permafrost base, slowed considerably when entering the domain of the silt with excess ice, the formation of a talik, and surface lowering in response to the loss of excess ice at the permafrost base and the permafrost table (Fig. 8).

During this simulation, the ice-rich permafrost of more than $60$ m thickness becomes near-isothermal, only minimally below $0\,°C$, with strongest loss of ground ice occurring at the permafrost table and the permafrost base where the temperature-depth profile is most strongly curved (Fig. B6). While this is a simplified case study, ice-rich permafrost tens of metres thick exists (e.g., Kokelj et al., 2017; Subedi et al., 2020; Young et al., 2022) and can be expected to undergo conditions of partial (simulation years $1,500$–$2,000$ in Fig. 8) or accelerated (after simulation year $2,000$) and eventually complete degradation under sustained anthropogenic climate change. While the choice of $80\%$ excess $100$ m thick may be extreme, this case study nevertheless illustrates that decaying bodies of ice-rich permafrost can persist under terrain surfaces without near-surface permafrost for centuries to millennia. During this time, the ongoing melt of ground ice may continue to condition hazards.





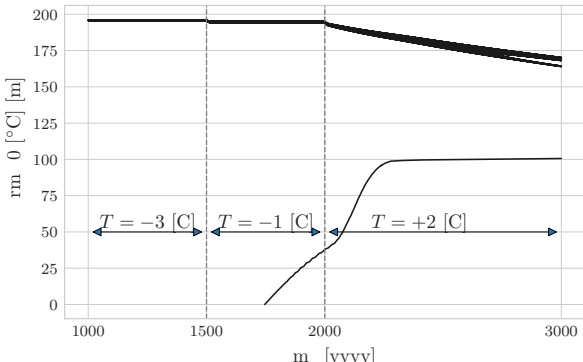

**Figure 8.** Position of the $0\ ^\circ$C isotherms over time. In the first period there is only one isotherm that oscillates with the surface boundary condition; the lower model boundary is below $0\ ^\circ$C. In the second period, the oscillating isotherm close to the surface is displaced with the subsiding ground (Fig. B4) and the rising permafrost base enters the model domain. In the third period, the permafrost base moves upward, time-lagged with respect the changing boundary condition. It stagnates near the interface between the granite and the silt with excess ice near 100 m, indicating the melt of excess ice at that depth. Close to the surface, the $0\ ^\circ$C isotherm reflects seasonal freezing and thawing, and a second isotherm emerges at the permafrost table, indicating the formation of a supra-permafrost talik (Fig. B4).

## 3.4 Case 3: Erosion and deposition

In this section we present two simulations in which we consider the deposition (Case 3D) and erosion (Case 3E) at the ground surface. We aim to demonstrate the ability to account for changes in geometry via boundary conditions as encountered for example in tailings deposition (Knutsson et al., 2018). Here, deposition and erosion at the surface affect the heat balance and hydro-mechanical aspects are disregarded.

Case 3D is like Case 2 and additionally includes depositions of $1\ \mathrm{mm\,yr}^{-1}$ of silty soil at the ground surface, with the same thermal properties as the autochthonous soil (Tab. 1), whereas Case 3E includes erosion of $1\ \mathrm{mm\,yr}^{-1}$ at the ground surface.

As for Case 2, both for Case 3D and Case 3E the bottom boundary condition is a constant geothermal flux, $0.05\ \mathrm{W\,m}^{-2}$, and at the surface, we impose a Dirichlet boundary condition as a sine wave with a mean temperature of $-3\ [^\circ\mathrm{C}]$ and an annual amplitude of $10\ [^\circ\mathrm{C}]$. The shallow spin-up is $15\ [\mathrm{m}]$ deep with a uniform initial condition $T = -3$

Figure (9) shows a comparison of the evolution of the surface elevation and the total volume of excess ice among Case 2, Case 3D, and Case 3E.

In the first 500 years, with boundary conditions equal to those during spin-up, we observe contrasting responses in ground surface elevation and excess ice loss. In Case 3D, surface elevation increased and the total volume of excess ice remained constant; whereas in Case 3E, surface elevation lowered although the column is in equilibrium with the thermal boundary conditions: the erosion disrupts the thermal equilibrium causing the thawing front to penetrate into the material with excess ice (Fig. 9b).




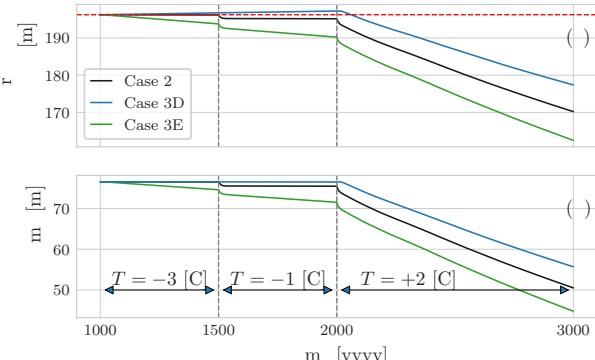

**Figure 9.** Panel (a) shows the time evolution of the ground surface elevation for Case 2 (reference), Case 3D (deposition of $1\,\mathrm{mm\,yr^{-1}}$), and Case 3E (erosion of $1\,\mathrm{mm\,yr^{-1}}$). The red dashed line indicates the surface elevation at the beginning of the simulation. Panel (b) shows the time evolution of the total volume of excess ice in the soil column for the three cases.

In the subsequent $500$ years, in Case 3D the is no melt of excess ice in contrast to Case 2 where the warming of the surface boundary condition caused a rapid thaw settlement. In Case 3D, the deposited soil prevents the thawing front from reaching the ice-rich layer. In Case 3E, the increase in surface temperature causes an increase in the average lowering rate of the ground surface from $4.9\mathrm{mm\,y^{-1}}$ to $7.1\mathrm{mm\,y^{-1}}$.

During the last $1,000$ simulation years, with a mean surface temperature of $2\,°\mathrm{C}$, thaw settlement occurs in all three cases but with different rates and timing. In fact, while in Case 2 and in Case 3E the increase in ground surface temperature determines a simultaneous loss of excess ice to which corresponds a lowering on the ground surface, in Case 3D the deposited soil delays the onset of excess-ice loss by 16 years (Fig. C1). The net effect of deposited volume of soil is to reduce the thermal gradient, thus the heat flux, between the ground surface and the top of the ice-rich layer. In Case 2 the average lowering rate of the ground surface is $24.9\mathrm{mm\,y^{-1}}$, in Case 3D $19.8\mathrm{mm\,y^{-1}}$, and in Case 3E $27.7\mathrm{mm\,y^{-1}}$.

The thaw response to increasing surface temperature differs between the cases with an without deposition (Fig. 10). With deposition (Case 3D), active layer thickness (ALT) and thaw penetration (TP) increase and quickly reach a new stable condition. This is because the active-layer thickening occurs in materials without excess ice. Without deposition (Case 2), the thickening of the active layer occurs at a slower rate due to the presence of excess ice (Shur et al., 2005; O'Neill et al., 2019). However, the melting of excess ice is manifested in larger TP and thaw settlement.

The thaw response differs from Case 2 (stable surface) and Case 3D (deposition) (Fig. 11). ALT decreases over time even though the thermal boundary conditions are constant. This occurs due to the melt of excess ice, also manifested in the trends of thaw penetration and the ground-surface elevation.

Figure 12 shows a comparison of the ground temperature at $10\,\mathrm{m}$ below ground surface for the Case 2, Case 3D and Case 3E. Figure 12a shows a comparison at the first surface temperature increase: in all three simulation the ground temperature presents a positive trend, more evident for Case 3D, whereas Case 2 and Case 3E present almost the same ground temperature,





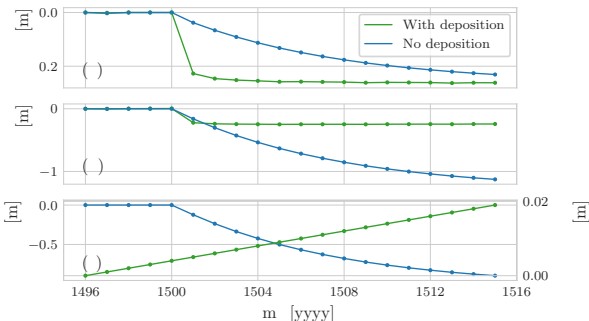

**Figure 10.** Case 3D and Case 2 produce contrasting thaw responses, illustrated for simulations years 1495–1515 for (a) active layer thickness (ALT), (b) thaw penetration (TP), and (c) elevation of the ground surface (GS).

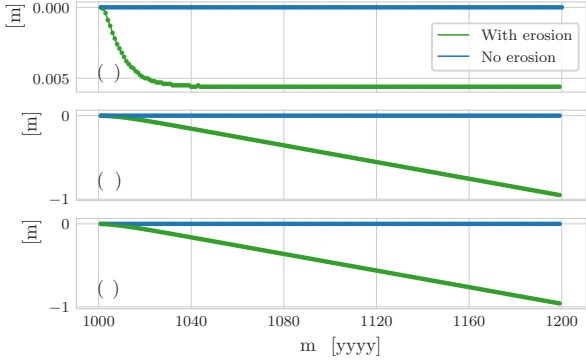

**Figure 11.** Case 3E (erosion) and Case 2 (no deposition) produce contrasting thaw responses in (a) active-layer thickness (ALT), (b) thaw penetration (TP), and (c) elevation of the ground surface (GS), shown here for simulations years 1000–1200.

and a reduction in the amplitude of the yearly oscillation due to the fact the deepening of the thaw penetration front occurs in ice-rich soil layers.

## 4 Discussion and conclusions

We have presented FreeThawXice1D as an extension of FreeThaw1D (Tubini et al., 2021), aiming to provide an accurate tool for research and to better understand the trade-offs involved in simulating the depth, timing, and magnitude of permafrost thaw in heterogeneous and ice-rich ground.

The model has six key capabilities: (1) Guaranteed convergence and conservation of energy even at large time steps enable millennium-scale simulations, even in the presence of steep enthalpy curves typical for excess ice. (2) A constitutive model allowing the gradual loss of excess ice from a control volume and the representation of layered and suspended cryostructures. Although not investigated here, the model in principle allows for the formation of ground ice when water transport is also mod-




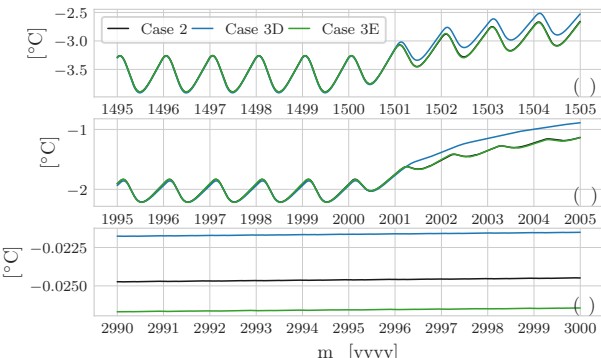

**Figure 12.** Ground temperature below 10 m the ground surface: panel (a) at the first surface temperature increase, panel (b) at the second surface temperature increase, and panel (c) at the last 10 years of the simulation.

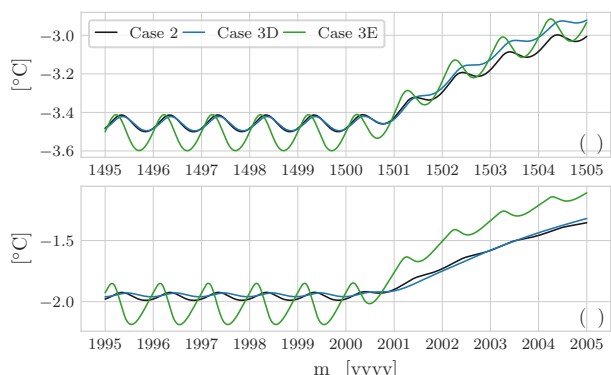

**Figure 13.** Ground temperature above 180 m the bottom column: panel (a) at the first surface temperature increase, panel (b) at the second surface temperature increase, and panel (c) at the last 10 years of the simulation.

eled. (3) Adaptive vertical discretization improving the tracking of $0\,^{\circ}$C isotherms and accommodating the melting of excess ice, as well as the deposition and erosion of material on the surface of the ground. (4) An efficient spin-up procedure allowing
to simulate deep soil columns accurately in terms of temperature, ice content, and a suitable geometry of the computational grid. (5) A parsimonious new boundary condition for representing seasonal variations in surface cover with a combination of time-varying heat-transfer coefficients and the enforcement of temperatures $\leq 0\,^{\circ}$C underneath snow. (6) Reporting of output quantities relative to a fixed datum and relative to a moving soil surface to enable the representation of differing modes of field observation.
The experiments presented demonstrate the importance of suitable numerical methods and solvers (cf., Tubini et al., 2021), dynamic and locally refined spatial discretization, and detailed representation of stratigraphy for accurately simulating the



depth, timing, and magnitude of ground-ice loss. Relaxing these requirements may result in output that reproduces long-term behavior but errs on details that may be important for investigating processes or informing forward-looking adaptation.

Combining an initial shallow spin-up with subsequent downward iteration of the equilibrium profile to depth has been shown
to be suitable for deep simulations. Testing spin-up completeness by comparing two separate simulations, one initialized at $-1$ °C and one at $+1$ °C, has shown to be a robust criterion in the absence of excess ice and water transport. By contrast, neither visual assessment nor seemingly sufficient (20,000 years) spin-up durations reliably produce identical results (Fig. 6). Having tested the two-step spin-up procedure in such a way, it can be confidently applied in simulations with excess ice. During spin-up, where melt and regridding may occur at the permafrost table and base, bringing the initial, potential cryostratigraphy,
surface elevation, and computational grid into agreement with the simulated ground thermal regime. This ability is a key for intersecting potential ground-ice distributions and profiles derived from geologic methods (Castagner et al., 2023; O'Neill et al., 2019) with simulated ground thermal regimes (Cao et al., 2019; Fiddes et al., 2015) that can add fine-scale spatial differentiation.

Deposition and erosion at the soil surface affect simulation geometry and can be a source of excess ice. Especially in
long-term simulations, they can affect the magnitude and timing of ice melt in the ground and FreeThawXice1D will allow simulation experiment for studying corresponding phenomena.

A parsimonious boundary condition for representing the seasonal variation in atmosphere-ground heat transfer has been developed and included in FreeThawXice1D. Its parameterization and testing are reported in (Brown and Gruber, 2025). This boundary condition overcomes the need for sub-diurnal temporal resolution and the data hunger of surface energy-balance
models while allowing us to take full advantage of the capability of using large time steps.

Depending on the task at hand, one may be interested in changes occurring at a depth measured relative to the ground surface or to a fixed datum. This becomes relevant when the surface elevation changes in response to excess-ice dynamics, erosion, or deposition (Fig. 8, 12, and 13). This issue has been recognized in field monitoring of active-layer thickness and borehole temperatures (Farquharson et al., 2019) and will become increasingly important with future climate change. FreeThawXice1D
provides an opportunity to investigate the long-term impact of differing observation modalities in simulation experiments.

FreeThawXice1D is a practical tool and stepping stone in further research. In the absence of analytical models that can be used as a reference, and it provides a possible benchmark for measuring how necessary trade-offs in other models of ground with excess ice affect simulation accuracy.

*Code and data availability.*  https://zenodo.org/records/15569126 The materials contain the source code and examples with Jupyter note-
books as model documentation and 'learn-by-doing' tools (Pianosi et al., 2020; Peñuela et al., 2021). They can be divided in two groups (Peñuela et al., 2021): Knowledge transfer model mainly aimed to present the input and output data together the relevant Python scripts, and Implementation Notebook meant to show how to set-up and use FreeThawXice1D.



## Appendix A: Parameterization of snow cover

Figures A1 and A2 provide additional insight into the simple snow-cover parameterization. Testing and parameterization are
reported in (Brown and Gruber, 2025).

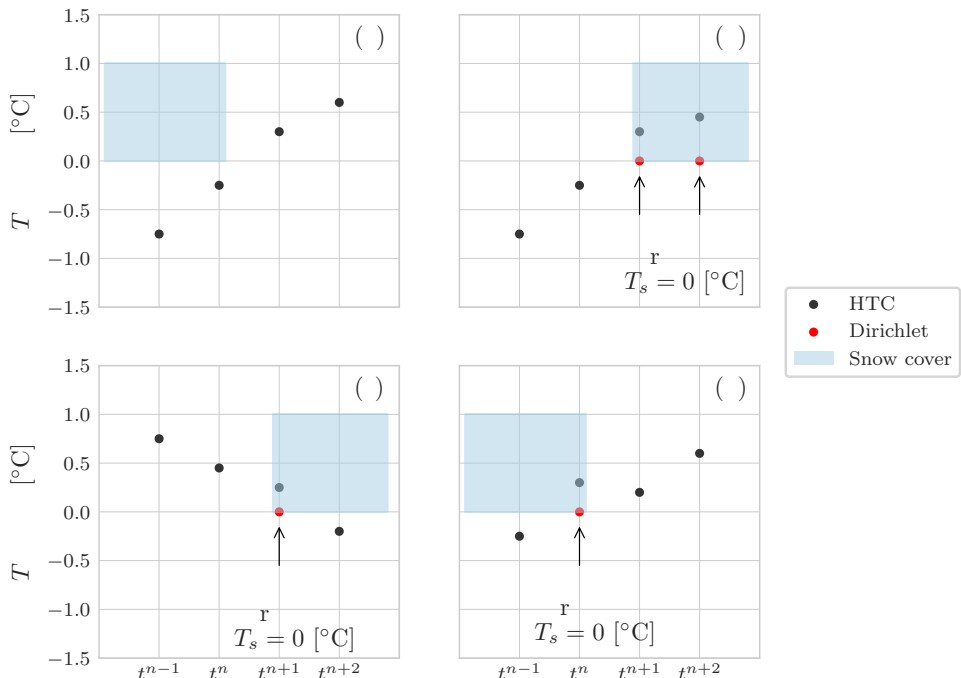

**Figure A1.** Schema illustrating the heat-transfer coefficient surface boundary condition combined with snow cover. The axes show the time step of numerical integration (horizontal) and the temperature of the uppermost control volume (vertical). Grey dots represent the numerical solution obtained with the heat-transfer coefficient (HTC) and red dots those with a Dirichlet boundary condition of $T_s = 0\,°$C. In (a), all the time steps are solved with the HTC boundary condition. In (b), time step $t^{n+1}$ and $t^{n+2}$ are solved twice: first with the HTC and then, because the resulting temperature is above $0\,°$C and there is snow cover, these two time steps are solved again with the Dirichlet boundary condition. In (c), the computation for time step $t^{n+1}$ is repeated by using a Dirichlet boundary condition, while the subsequent step is adequately solved with the HTC condition as the resulting temperature $T_{KMAX}$ is below $= 0\,°$C. In (d), the computation for time step $t^n$ is repeated switching the boundary condition, whereas $t^{n+1}$ is adequately solved with the HTC condition as there is no snow cover.





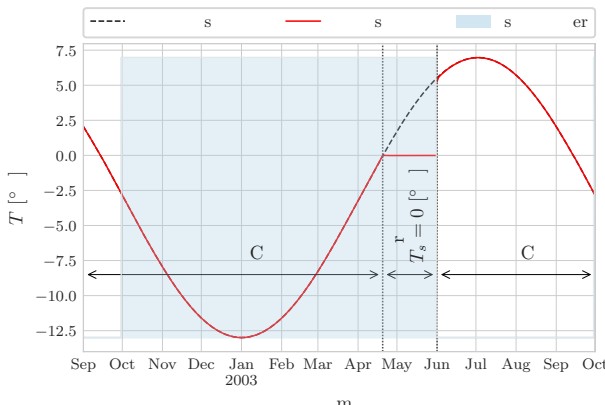

**Figure A2.** Behavioural simulations demonstrating the heat-transfer coefficient (HTC) boundary condition with and without snow cover. The vertical axis shows the temperature of the uppermost control volume. The red line represents the temperature computed by switching the surface boundary condition to Dirichlet with a constant temperature of 0 °C in the presence of snow, whereas the black dashed line represents a simulation without switching the surface boundary condition in response to snow. The dotted vertical lines indicate when the boundary condition is switched.

## Appendix B: Case 2

Figures B1 B2, B3, B4 and B6 provide additional detail for the interpretation of Case 2.

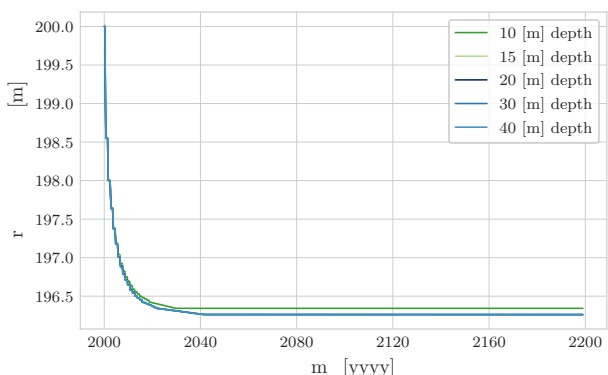

**Figure B1.** Surface elevation demonstrates the equilibration of near-surface excess ice with the boundary conditions during spin-up. Equilibration quality can be reduce if the initial ice loss makes the chosen domain shallower that the depth of relevant temperature variation during the spin-up period.



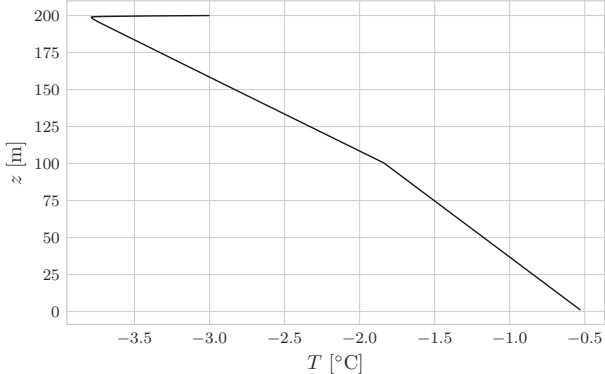

**Figure B2.** Equilibrium temperature profile for Case 2 obtained with the two-step procedure. Around $z = 100$ m, the temperature gradient changes at the boundary of silt and granite.

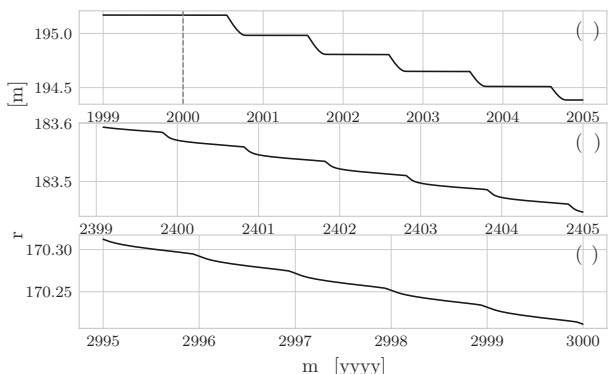

**Figure B3.** The seasonal pattern of surface subsidence during the last warming period with a mean surface temperature $+2\,^{\circ}$C. The seasonal pattern is diminishing over time as the permafrost table is located progressively deeper in the ground.



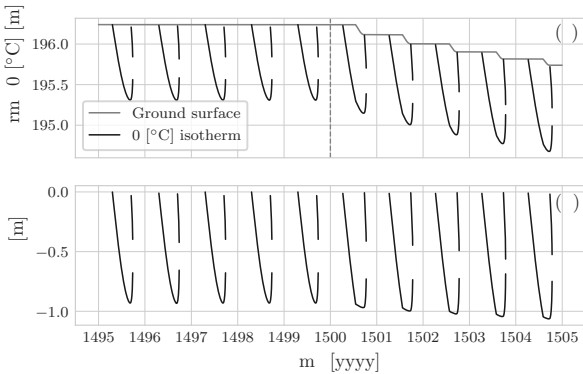

**Figure B4.** The position of the 0 °C isotherm around the time when the rise in mean surface temperature to $-1$ °C occurs and the ground surface subsides due to the loss of excess ice. In the presence of surface subsidence, it matters whether thaw depth is expressed relative to a fixed datum (a) or the moving ground surface (b). Expressed as annual extremes, only the former is a reliable measure of permafrost degradation at the permafrost table (thaw penetration), whereas the latter presents a low-biased signal. The steps reflect that subsidence here occurs only during the warm season and lags the 0 °C since the signal of the boundary condition because it takes time to reach the depth where excess ice is present.

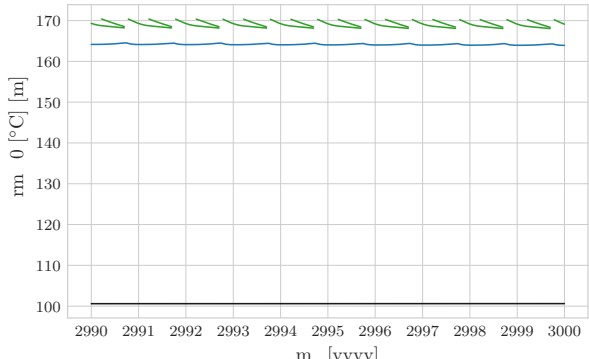

**Figure B5.** Details of the position of the 0 °C isotherm in the last 10 years of the simulation. The black line is the lowermost isotherm and indicates the permafrost base, the blue line represents the permafrost table, and the green line corresponds to seasonal freezing above a supra-permafrost talik.



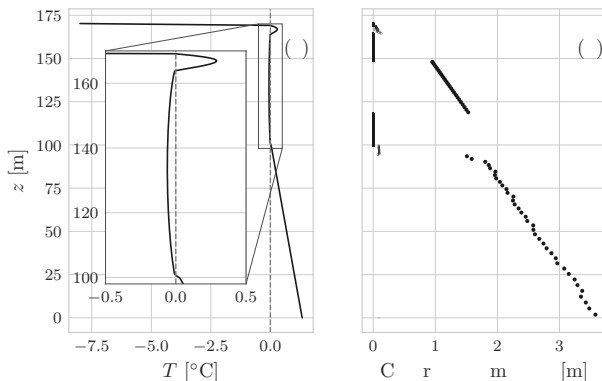

**Figure B6.** Near-isothermal conditions during permafrost thaw: Panel (a) shows the temperature profile at the end of the simulation, near-isothermal close to 0 °C between 100 m and 170 m. Panel (b) shows the control volume size for the last time step of the simulation.

## Appendix C: Case 3

Figure C1 provides additional detail for the interpretation of Case 3.

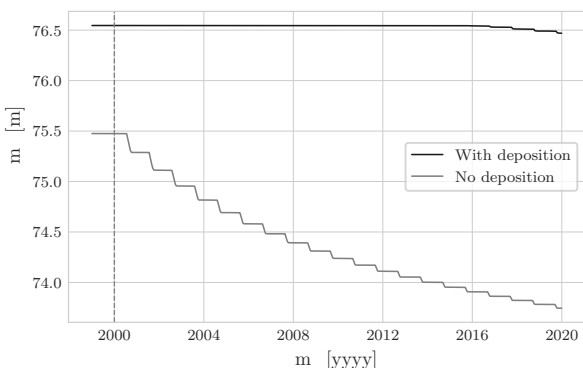

**Figure C1.** Geometry and geometry changes affect the timing and magnitude of ice loss in the ground, as demonstrated by the comparison between Case 2 (no deposition) and Case 3D (deposition). The vertical grey dashed line indicates the time when the mean surface temperature increases to 2 °C. Without deposition, the excess-ice melt occurs quickly as the temperature increases, whereas with deposition, there is a lag of almost 16 years.

*Author contributions.*  SG and NT conceptualized the study; NT developed the software and wrote the original draft; SG and NT reviewed and edited the manuscript.



*Competing interests.* The authors declare that they have no conflict of interest.

*Acknowledgements.* This research has been supported by a PhD grant by the Department of Civil, Environmental and Mechanical Engineering at the University of Trento (Department of Excellence of MIUR). In Canada, support was available through the NSERC Strategic

Project "Improved Characterization of Permafrost Vulnerability to Support Decision Makers, Infrastructure, and Community Stewardship in the Northwest Territories" (STPGP 521584) and NSERC PermafrostNet (NETGP 523228-18). The Digital Research Alliance of Canada provided high-performance computing resources via RPP 772.



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
