# Peer review of "Simulating soil heat transfer with excess ice, erosion and deposition, guaranteed energy conservation, adaptive mesh refinement, and accurate spin-up (FreeThawXice1D)"

_EGUsphere, 2025_

## Referee Comment (RC1)

Comment on egusphere-2025-2649

**Overall comment**

This manuscript presents a new model scheme to simulate the thawing of ice-rich permafrost. It is a relevant and up-to-date topic, particularly as subsidence following excess ice melt can have impacts on geomorphological features and the stability of infrastructure. The study presents a model approach with a high computational efficiency, allowing the simulations of long time spans, making it an interesting tool for the scientific community.

However, the model description is lacking important information to understand the approach and uncertainties as well as limitations of the model are not discussed. Furthermore, a model validation and/or real-world application is missing, giving the impression of a model development paper. In this case, the benefits of this approach should be better highlighted.

Major corrections should be implemented before publication.

**Major comments**

The authors present a model description where the most important functionalities of the code are explained. However, some parts are difficult to understand due to missing information. Particularly the following points should be clarified and described in more detail before publication:

-   The surface boundary conditions are unclear, or rather how this is physically reasonable, especially with the representation of the snow cover (see point below).
-   What about lateral boundary conditions?
-   The snow-melt isotherm is in many locations not the dominant influence of the snowpack on the ground thermal regime, but the thermal properties of the snow, which have a distinct influence on the ground temperatures. Therefore, the approach used in the presented model is a simplification, and it is important to improve the clarity of this approach in the main part of the paper and not only in the appendix. A justification why this approach was chosen should be added.
-   It is unclear how the temperature threshold was chosen for the grid refinement. This should be dependent on the soil type / freezing curve and should be tested with a sensitivity test in the results. A discretization close to the zero-isotherm might not be the best choice depending on the material (ice/soil/mineral mixture).
-   The authors state that the model can represent the formation of new excess ice (L. 65, 152, 175, 377). This statement should be deleted or the process of excess ice formation should be described in the model description. However, it is unclear how excess ice formation can be simulated when water fluxes are neglected.
-   It is not described in the methods how erosion and deposition is handled in the model. Is the material added/removed once a year? And where do those information come from? It is challenging to find long-term

sedimentation rates, and also for shorter time spans, it is challenging to find a representative setting, as deposition typically varies in time and magnitude. See also third major comment.

It is strongly recommended to include a new section describing the uncertainties of the model. Uncertainties, limitations and simplifications are inevitable when simulating excess ice thaw over long timescales. However, they have to be critically discussed. Particularly the following points should be covered, but other aspects might also be important to mention.

- 1D model setup
- Applied boundary conditions
- Snow representation: As far as we understood, the snow cover in the model only suppresses positive ground temperatures, and does not affect them as long as they are below 0°C. This is an important uncertainty, as a snow cover considerably influences also sub-zero ground temperatures depending on the snow thickness and properties.
- Not considering other influences on the ground thermal regime: What about vegetation, evapotranspiration and other surface energy balance fluxes?
- Neglecting water fluxes: The model immediately removes the melt water. Discuss the uncertainties connected to this simplification.
- Neglecting excess ice formation: The model also neglects the formation of new excess ice, e.g. through ice segregation. This can be especially important over long timescales and under deposition and can decrease the total amount of subsidence. In case excess ice formation is included in the model (unclear if that is like that, see first major comment), it has to be described in the methods.

The study gives the impression that the authors present a model development paper as idealized cases are presented and no applications of the model are presented. It is recommended that the authors clearly state that the aim is a new model development and include the following points in their discussion:

- Clearly discuss the benefits of their new model approach, particularly regarding spinup, grid refinement and computational time for long term runs.
- The authors present several test cases and compare their results against each other. The setup of all these cases is synthetic and a model validation with data from the field or laboratory tests is not performed. Therefore, it is not possible to assess the model performance and how well it can represent real conditions. It is recommended to include some form of model validation if possible.
- It would be beneficial for the discussion to include a paragraph about the applicability of the model on real cases. As the authors mainly demonstrate the capability of the model to run long-term simulations: Is it realistic to apply the model as it is to a field site? What would be the challenges, e.g. available climate data, available sedimentation/erosion rates, available data to initialize the ground ice content?

**Minor comments**

All figures should be checked concerning the labeling of the y-axis and the numbering of the sub plots. Furthermore, there are random letters (r, rm, er, m) in the figures which should be removed or explained.

Line 7: What is meant with soil geometry? Do the authors refer to surface geometry as in line 14?

Line 28-30: Improve the clarity of the sentence, especially the word "intersection" does not fit very well.

Line 42: Update the reference, this model is now called "CryoGrid community model" and the newest reference is Westermann et al., 2023 (https://doi.org/10.5194/gmd-16-2607-2023)

Line 59-74: Missing a discussion on model errors/uncertainties due to spinup, the build-up of excess ice and accurate model forcing during ice formation/spinup and the time period of interest with excess ice melt. How does the gained accuracy by this scheme compare to uncertainties in spinup?

Line 61: "The duration and depth are connected" is unclear.

Line 69: "The spinup of soil profiles into equilibrium accommodates the interacting changes in temperature, geometry and ice content": this implies that the formation of excess ice is described.

Line 81: The source-sink term S should be explained in more detail. What sources and sinks does the model account for?

Line 88 / equation 3: Why (1 - theta(s))? Why not theta(sp)?

Line 94: When using SFCC the first time in the methods, write it out.

Line 87 / equation 6: define subscript "ei".

Line 101: How is the temperature range parametrized for different soil types? Based on the SFCC?

Line 118: add reference

Line 130: How does one know which mean to use if there is no sample of the soil and the cryostructures are unknown? Is there a default in the model?

Line 133: typo: model → models

Line 136: Give a (very) short summary of the Johansen model and the Cosenza model for readers who are not familiar with them.

Line 149: "Excess ice content and geometry are directly linked": Unclear, stating the obvious, how does it play out here? Equation or example?

Section 2.5: Generally more description in this section, this is very important, maybe a plot.

Section 2.8: Unclear, from the description it seems that the post-spinup "equilibrium" will be sensitive to the threshold between the two different spinup

methods. It would be good to see a sensitivity test there or show the timeseries. Again, show why the choice around the zero-isotherm is reasonable here and how wide the refinement area is, this seems very relevant.

Section 2.9: Where does the input come from?

Line 259: What soil type was used for this test case and where does this data come from? It affects the SFCC and the contents of soil particles, water and ice in the soil. Furthermore, it would help the reader to judge, if the subsidence is in a reasonable order of magnitude. Would be good to have a real case study here.

Line 260: The starting conditions of this test case seem unrealistic. You assume a temperature gradient of 5 °C over 10 m. Which soil type do you want to represent with this setting (see comment above)? Find a more realistic setting or give a reasoning for it.

Figure 3: This figure needs to be improved: Numbering a, b and c is missing. Y-axis should be labelled as "surface position" for a, and "surface elevation change" for b and c or something similar. It is very difficult to distinguish the colours with are not dark green and red.

Section 3.1: It is clear that different grids will change the signal and that they will converge at the higher resolution. However, a validation against a real case is missing.

Figure 4: This figure needs to be improved: y-axis should be labelled as "surface position" or something similar.

Table 1: State the source for the van Genuchten parameters. For example, Dall'Amico (a reference that the authors cite) give a values of 0.65 m and 1.67, which differ considerable from the values presented here.

Table 1: State the depths of silt and rock that are given in line 298.

Table 1: What is theta(s) and theta(r)? In the methods the authors use different subscripts. It would be helpful to give all abbrevations in the table caption.

Figure 5: Abb a and b to the figure. And what is "r" and "rm"? These letters were also appearing in Figure 3 and 4 before, and were not explained.

Line 295: Put this explanation at the beginning of the section.

Line 298: The authors explain their reasoning for their extreme model setup with 80% excess ice until 100m depth in line 327 ff. However, a more realistic setup of the model is recommended.

Line 303: The authors state that the shallow domain should be chosen sufficiently deep. Which depth did they chose?

Line 322: Explain where the 60 m of ice-rich permafrost comes from.

Line 343: In the case of deposition, typically new excess ice would be formed through segregation, resulting in an even more pronounced surface elevation rise. Discuss this in the uncertainties.

Line 362: Figure 11 shows the case 2 (stable surface) and Case 3E (erosion). Check your description in lines 362-364.

---

## Referee Comment (RC2)

This review is conducted as a joint effort within the EGU peer review mentoring program under the mentorship of the TC editor Hanna Lee. The review provided below is a collective summary from myself as well as the trainees of the program.

This study describes FreeThawXice1D, a one-dimensional permafrost heat-transfer model designed to simulate excess ice melting and subsequent ground subsidence over centuries to millennia while conserving energy and remaining numerically stable with large time steps. The authors focus specifically on improving spin-up to achieve thermal equilibrium, adaptive mesh refinement, flexible boundary condition that approximates snow and surface heat exchange without full energy-balance forcing by setting up idealized test cases. The study is primarily a numerical methods and model-capability demonstration rather than representing real-world permafrost thaw processes, highlighting how accurate spin-up and dynamic regridding are crucial for representing deep ice-rich permafrost thaw.

The manuscript presents an innovative and technically strong open-source release 1-D permafrost model (FreeThawXice1D) with energy conservation, adaptive meshing, and efficient spin-up. While the manuscript focuses on technical details of the numerical methods, it fails to clearly describe the novelty of this study and the scientific purpose of the model development. At this time, the manuscript reads more like a technical description of a model and not a scientific paper enough to attract broader audience interested in permafrost processes and modeling. A stronger contextualization of the main purpose, validation or comparison, better figures and structure, and an explicit limitations section is warranted to guide users on when the model is (and isn't) applicable.

Below are some of the main concerns that need to be addressed.

The spin-up case is highly idealized using one type of soil and large amounts of excess ice. It is not clear how realistic this case set up is. To demonstrate and increase generality, the authors should conduct at least one or two contrasting soil setups, e.g.

- coarse sand/gravel with low heat capacity and low excess ice
- peat or organic-rich soil with high heat capacity and high unfrozen water content
- a moderate ice-content silt (30–50 %)
- comparison of existing data or literature may also be necessary to demonstrate whether the model results are reasonable or not
- sensitivity test

This could be small additional runs enough to show that the model is robust beyond one synthetic silt column with large amounts of excess ice.

Apart from not representing various realistic cases of soil columns, the model has several limitations. It is difficult to understand what the application of this model could

be as the study does not show climatic responses of permafrost. In addition, hydrological process representation is missing in this 1D model, thus it is unclear what the fate of water after excess ice melting will affect additional heat balance change associated with excess ice melting, which could be a large energy transfer mechanisms in permafrost soils. This may be beyond the scope of this study; however, such large limitations are largely ignored in the discussion section. The authors claim 'FreeThawXice1D is a practical tool and stepping stone in further research.' But I struggle to understand the practicality of this model that the future users outside of this model developer group can use this model for. I suggest including model limitations and future work paragraph in discussion section. Also, the authors can provide some examples of how this model can be used either in introduction or discussion section.

The current version of the manuscript is quite figure heavy and the figures are not necessarily very clear in their purpose, therefore, they appear quite redundant. Additionally, the figures are not very aesthetically presented (e.g. captions don't fully explain what's shown, missing or inconsistent panel labels, confusing units (mm vs m), and hard-to-read colors). In my opinion, the main paper could cut down the number of figures from 14 to 7-8 with concise composite figures with clear purpose. For instance, figures 7 & 8 may be merged, figures 10 & 11 could be combined as split panels in one figure, and figures 12 & 13 can be more simplified and be more explicit. I suggest the authors to thoroughly check the captions and legends during the revision and increase reader friendliness of the figures.

Below are line by line comments.

Line 50: While it is acknowledged that existing models have limitations, a more detailed exploration of these shortcomings is crucial for underscoring the relevance of this study. Merely stating that limitations exist does not adequately convey the need for developing a new model. Please provide a better contextual background on what is specifically lacking in the existing models and why the new model developed is needed.

Line59: Can you be more specific about in what context this is more desirable?

Line64: It is mentioned in the introduction that the work builds on previous work by Tubini et al. 2021, but there is no mention of that it is an extension of the FreeThaw1D model until line 370. Only then it is suggested that the model already existed, and only new features were added. It is unclear in the introduction of the six aspects are mentioned are new specifically to this model, were already part of the previous model. To enhance clarity, it is recommended that the introduction be revised to explicitly state that this work is an extension of the FreeThaw1D model, rather than a wholly novel development; with clear descriptions of which aspects already existed and what is the novelty in this specific model. This could be further expanded within the methodology where currently the full model is described as new / implementing other methods for

the different aspects. At this time, the purpose of the model development or extension is not clear. Please describe the general model framework and the purposes of this model use.

Section 2.5 Excess ice: It appears that excess ice only melts in this model as well. So why simulating thousands of years is important and necessary when it will only melt?

Section 2.7 Regridding: I am very confused why regridding is necessary in a 1D model. Could you elaborate?

Section 3.1: It is very difficult to understand what experiments have been conducted for what purpose. Is it simply to compare the regridding effects? If so, regridding effects on what aspects of the model representing permafrost thaw processes? Please add more description to help the readers.

Line298: How likely is this scenario? Can you reference a dataset or a study showing how plausible this scenario is?